# Dimethyl Fumarate and Intestine: From Main Suspect to Potential Ally against Gut Disorders

**DOI:** 10.3390/ijms24129912

**Published:** 2023-06-08

**Authors:** Federico Manai, Lisa Zanoletti, Davide Arfini, Simone Giorgio De Micco, Arolda Gjyzeli, Sergio Comincini, Marialaura Amadio

**Affiliations:** 1Department of Biology and Biotechnology “L. Spallanzani”, University of Pavia, 27100 Pavia, Italy; federico.manai01@universitadipavia.it (F.M.); lisa.zanoletti01@universitadipavia.it (L.Z.); davide.arfini01@universitadipavia.it (D.A.); simonegiorgio.demicco01@universitadipavia.it (S.G.D.M.); arolda.gjyzeli01@universitadipavia.it (A.G.); sergio.comincini@unipv.it (S.C.); 2Department of Chronic Diseases and Metabolism (CHROMETA), Katholieke Universiteit Leuven, 3000 Leuven, Belgium; 3Department of Drug Sciences, University of Pavia, 27100 Pavia, Italy

**Keywords:** dimethyl fumarate, intestine, inflammation, inflammatory bowel diseases, Crohn’s disease, ulcerative colitis, celiac disease, gut disorders, Nrf2, repurposing

## Abstract

Dimethyl fumarate (DMF) is a well-characterized molecule that exhibits immuno-modulatory, anti-inflammatory, and antioxidant properties and that is currently approved for the treatment of psoriasis and multiple sclerosis. Due to its Nrf2-dependent and independent mechanisms of action, DMF has a therapeutic potential much broader than expected. In this comprehensive review, we discuss the state-of-the-art and future perspectives regarding the potential repurposing of DMF in the context of chronic inflammatory diseases of the intestine, such as inflammatory bowel disorders (i.e., Crohn’s disease and ulcerative colitis) and celiac disease. DMF’s mechanisms of action, as well as an exhaustive analysis of the in vitro/in vivo evidence of its beneficial effects on the intestine and the gut microbiota, together with observational studies on multiple sclerosis patients, are here reported. Based on the collected evidence, we highlight the new potential applications of this molecule in the context of inflammatory and immune-mediated intestinal diseases.

## 1. Dimethyl Fumarate Pharmacokinetics and Mechanisms of Action

Dimethyl fumarate (DMF; IUPAC name: dimethyl (*E*′)-but-2enedioate) is an α,β-unsaturated carboxylic ester with a molecular weight of 144.13 g/mol derived from fumarate (or fumaric acid, FA), an organic compound found in *Fumaria officinalis*, *Boletus fomentarius*, and *Cetraria islandica*. The synthesis of DMF can be performed following two different approaches: one based on the isomerization of dimethyl maleate through reversible addition of an amine catalyzed by a Lewis acid, and another involving a Fisher esterification of FA. DMF and its active metabolite monomethyl fumarate (MMF) exert a wide range of biological functions, in particular as anti-inflammatory and immune-modulatory agents [1,2,3,4,5,6]. According to its high tolerability as well as its well-studied mechanisms of action, DMF is currently approved by the US Food and Drug Administration (FDA) as the commercial products Fumaderm^®^ and Tacfidera^®^ for the treatment of psoriasis and relapsing-remitting multiple sclerosis (RRMS) [7,8,9,10].

The pharmacokinetics of DMF has been thoroughly characterized: specifically, after oral administration, DMF is normally undetectable in the bloodstream [11] and urine [12], while its main metabolite MMF is present and shows a systemic peak after 2.5–3.5 h [11,13]. In vitro studies clarified the turnover of DMF in the gastrointestinal (GI) tract and its pre-systemic metabolism. Particularly, in permeation experiments with intestinal mucosa, Werdenberg and colleagues [14] demonstrated that DMF was rapidly hydrolyzed when in contact with intestinal homogenates and perfusates compared with those derived from pancreatic tissues (used as control), due to the high concentration of carboxylesterases (present in perfusate) and carboxyl-/cholinesterases (in homogenate). Moreover, intestinal homogenates show a higher efficiency in DMF catabolism with respect to the perfusates. The reaction catalyzed by intestinal esterases in the lumen and mucosa generates, in a 1:1 ratio, MMF and methanol; the latter is then converted into formic acid by intestinal alcohol dehydrogenases (e.g., ADH1A and ADH4). The in situ levels of methanol and formic acid are the main factors responsible for DMF-related GI symptoms and ailments experienced by some patients (e.g., nausea, diarrhea, abdominal pain, cramps) [15]. The pharmacokinetics of DMF is also influenced by its instability in biological fluids and lipophilicity (10-fold higher vs. MMF), the latter likely explaining its rapid penetration in the tissues and its potent biological effects compared with the relative monoester [14]. In particular, permeability experiments with intestinal mucosa in an Ussing-type chamber further suggested that DMF is completely metabolized in the intestine, given that only MMF was detected in the lower chamber. Additionally, the environmental pH plays a role in DMF pharmacokinetics since this molecule is preferentially hydrolyzed at alkaline pH (as in the intestine) instead of acidic pH [16]. In vivo studies using adult male rats demonstrated that soon after DMF is given directly into the small intestine, it reacts with glutathione (GSH) to form the adduct *S*-(1,2-dimethoxycarbonylethyl)glutathione (GS-DMS), as confirmed by the absence of free DMF and the presence of MMF and GS-DMS in the plasma of the portal vein blood. Moreover, in the mucosa of the small intestine, DMF and MMF are mainly detected as adducts with GSH (i.e., GS-DMS and GS-MMS) [17]. Notably, in vitro evidence shows that DMF is hardly hydrolyzed to MMF in a buffer of pH 7.4, but it is rapidly hydrolyzed is human serum at the same pH, indicating the need for enzymatic reactions. More specifically, compared to serum, the half-life of DMF in whole blood samples is drastically reduced, mainly for the metabolization promoted by monocytes/lymphocytes, and also, the concentrations of MMF and monoethyl fumarate decrease over time [16]. Figure 1 shows a schematic representation of DMF’s fate in the intestine:

The main and best-studied mechanism of action of DMF/MMF relies on activation, at low concentrations, of the translocation of the nuclear factor erythroid 2 [NF-E2]-related factor 2 (Nrf2, Nfe2l2). At basal conditions, Nrf2 is retained in the cytoplasm, inhibited, and catabolized after ubiquitination and proteasomal degradation through the binding with its repressor, Kelch-like ECH-associated protein 1 (Keap1) [18]. Specifically, DMF’s mechanism of action is mainly based on the reaction of succination exerted on cysteine residues on Keap1, thus leading to Keap1 conformational change and release of Nrf2, although other binding sites within the Keap1 structure have been identified as contributors to DMF’s pharmacological activity [19,20]. The consequent translocation of Nrf2 into the nucleus promotes the transcription of target genes carrying in their promoter regions cis-regulatory enhancers, the so-called antioxidant response elements (AREs), which encode for cytoprotective and detoxifying enzymes. Among others, some Nrf2 target genes are *Heme Oxygenase-1* (*HO-1*), *NAD(P)H Quinone Dehydrogenase 1* (*HQO1*), and *γ-Glutamylcysteine Synthetase* (*GCLC*) [21,22,23] (Figure 2). Notably, Nrf2 can induce also the DNA damage signaling pathway through *TP53* transcription, whose promoter contains three AREs [24], thus connecting Nrf2 with the autophagy pathway, which is in turn regulated by p53 in oxidative stress contexts [25]. Importantly, in vivo experiments using a mouse model with *Atg5^−/−^* hepatocytes demonstrated that autophagy dysregulation leads to a prolonged Nrf2 activation as a compensatory mechanism [26]. However, the crosstalk between Nrf2 and autophagy mainly occurs through sequestosome1 (SQSTM1/p62). This autophagy adaptor protein promotes the autophagy-mediated degradation of Keap1, thus leading to the activation of Nrf2. The regulation of Keap1 cytoplasmic levels by SQSTM1/p62 has been demonstrated through both *SQSTM1/p62* overexpression and knockdown experiments [27]. The SQSTM1/p62-mediated degradation of Keap1 is also dependent on Sestrin1 and Sestrin2 [28]. The positive feedback loop between Nrf2 and autophagy is further strengthened by the regulation of the SQSTM1/p62 expression levels by Nrf2 itself [29]. *Nrf2* knockdown is accompanied by impaired SQSTM1/p62 levels and, similarly, *SQSTM1/p62* knockdown leads to a decrease in Nrf2, HO-1, and NQO1 levels [30]. In addition, in the presence of high reactive oxygen species (ROS) levels, 5′AMP-activated protein kinase (AMPK) also induces Nrf2 activation and sustains autophagy due to its negative effects on the mammalian target of rapamycin (mTOR); thus, as a regulatory triangle, AMPK, Nrf2 and mTOR guarantee a dynamic and precise control of autophagy under oxidative stress [31,32]. In agreement, AMPK can facilitate Nrf2 nuclear translocation [33] and boost the antioxidant Nrf2/HO-1 signaling pathway [34]. 

The pharmacological action of DMF is not limited to the Nrf2 pathway; it is performed also by Nrf2-indepenent mechanisms. For example, Nrf2 inhibits aerobic glycolysis through glyceraldehyde 3-phosphate dehydrogenase (GAPDH) succination in myeloid and lymphoid cells [35]; inhibition of the nuclear translocation of nuclear factor kappa-light-chain-enhancer of activated B cells (NF-κB) protein mediated by hydroxycarboxylic acid receptor 2 (HCAR2) represents another biological mechanism that contributes to the anti-inflammatory properties of DMF. Particularly, DMF-mediated HCAR2 activation leads to the induction of the AMPK-SIRT1 pathway, which promotes NF-κB deacetylation and, as a consequence, its inhibition [36,37,38,39]. Notably, some evidence suggests that DMF inhibits the expression and function of the hypoxia-inducible factor-1α (HIF-1α), by promoting its misfolding and consequently its proteasome-dependent degradation [40]. Indeed, DMF interferes with the interaction between HIF-1α and chaperone heat shock protein 90 (Hsp90) and promotes HIF-1α binding to the receptor for activated C kinase 1 (Rack1), which is able to induce HIF-1α degradation. Notably, a decrease in caspase-1 and NLR family pyrin domain containing 3 (NLRP3) has been described as another effect of DMF [41,42].

## 2. Dimethyl Fumarate and Intestine: State of the Art

Starting from 1994, the number of publications regarding the beneficial effects of DMF in intestinal tissues gradually increased. Indeed, gut inflammatory disorders (e.g., inflammatory bowel diseases and celiac disease) are still characterized by a high socio-economic burden and a consistent impact in the patient’s overall quality of life [43,44]. The number of entries in PubMed obtained by searching “Dimethyl fumarate AND intestine”, 40, is very similar to those of other tissues (e.g., heart = 44; eye = 26) and associated disorders, in which the repurposing of DMF has been already postulated [45,46,47,48,49,50,51]. In the following paragraphs, the state of the art regarding the effects exerted by DMF on intestine and gut microbiota will be highlighted. The search was performed on PubMed using “Dimethyl fumarate AND intestine” as keywords to isolate the relevant research articles focused on this topic. Additionally, synonyms of intestine (i.e., gut and bowel) were used. Reviews were excluded.

### 2.1. Dimethyl Fumarate’s Effects on Intestinal Mucosa

The first evidence regarding DMF’s effects in the intestine came from comparative studies with other molecules published starting from the mid 1990s. It was demonstrated that DMF was able to reduce the foci of aberrant crypts induced by azoxymethane in the colon of male Fisher rats [52]. Moreover, rats pre-treated with DMF and challenged with a toxic dose of naphthoquinones showed a higher increase in the DT-diaphorase levels in the intestine compared with those obtained with other inducers and, as a consequence, a higher amount of activated 2-hydroxy-1,4-naphthoquinone. In addition, pre-treatment with DMF decreased the renal lesions in these rats [53]. Several papers demonstrated that DMF exerted its beneficial effects in the intestinal tissue mainly through the activation of Nrf2 and inhibition of NF-κB. The Nrf2-dependent response and protective effects of DMF were first described in an aberrant crypt foci (ACF) model of colon carcinogenesis. As reported, DMF significantly induced the activity of both NQO1 and glutathione S-transferases (GST) in colon, the former being indicated by authors as relevant to inhibiting the initiation of carcinogenesis [54]. Another, more recent, study proved the anti-proliferative impact of DMF in vitro in human colon cancer cells by arresting the cell cycle in G0/G1 phase as well as activating both autophagy and apoptosis [55]. DMF can also promote colon cancer cell death through glutathione depletion and consequent apoptosis/necroptosis [56,57]. Using in vitro Caco-2 cells, another study demonstrated that DMF is able to protect against H_2_O_2_-caused barrier dysfunction by promoting HO-1-mediated zonulin-1 (ZO-1) induction [58].

In different in vivo models of intestinal disorders, DMF leads to the activation of Nrf2 and the downstream effectors, such as HO-1, GCLC, and GPX. Specifically, DMF induced an increase in GSH content as well as levels of Nrf2 and its target genes (i.e., HO-1 and HQO1) in a murine model of dextran sulphate sodium (DSS)-induced colitis [41]. The DMF-mediated increases in Nrf2 and HO-1 levels were also observed in the ileum of a rat model of experimental autoimmune neuritis [59] and in the intestine of an ischemia/reperfusion injury rat model [60], respectively. Additionally, oral administration of DMF in the DSS-induced colitis model led also to a reduction in weight loss, colon length shortening and tissue damage. Furthermore, DMF promoted an increase in GSH levels, as well as suppression of interleukin-1β (IL-1β), tumor necrosis factor-α (TNF-α), and IL-6 at both the mRNA and protein level. The described decrease in caspase-1 and NLR family pyrin domain containing 3 (NLRP3) seems to be caused by the reduction of mitochondrial ROS (mROS) and mitochondrial DNA (mtDNA) release [41], all consistent with DMF’s antioxidant properties. In the experimental autoimmune neuritis model, DMF was able to reduce mRNA levels of Toll-like receptor 4 (TLR4), interferon-γ (IFN-γ), and forkhead fox P3 (FoxP3) in the intestinal lamina propria. Moreover, the authors described an increase in CD4+CD25+ regulatory T-cells in Peyer’s patches [59]. Similar results were also obtained in the intestinal ischemia/reperfusion model, in which the DMF reduced the levels of IL-1β, TNF-α, MPO, iNOS, P-selectin, caspase-3, and glycogen synthase kinase-3β (GSK-3β). A decrease in NF-κB was also detected, suggesting a concomitant Nrf2-independent mechanism of DMF in this model [60]. Protection against hemorrhagic diarrhea and weight loss was demonstrated also in a model of colitis induced by dinitrobenzene sulphuric acid (DNBS). Again, DMF prevents the increase in MPO, TNF-α, and intercellular adhesion molecule-1 (ICAM-1) [58]. Similar outcomes were reported also in two additional studies, one based on DSS-induced colitis and the other one on necrotizing enterocolitis induced by hypoxia and lipopolysaccharide (LPS). Particularly, in the former study, DMF reduced the weight loss and the abdominal distension associated with diarrhea. The intestinal damage reduction relied mainly on the decrease in IL-1β, TNF-α, IL-6, NF-κB, TLR-4, Bcl-2-associated X protein (BAX), and effector caspases [61]. The latter study described the beneficial effects of DMF oral administration, which led to a reduction in intestinal length shortening and cyclooxygenase-2 (COX-2) levels as well as an increase in the antioxidant enzymes glutamate-cysteine ligase catalytic subunit (GCLC) and glutathione peroxidase (GPX) [62].

The Nrf2-independent beneficial effects of DMF were observed in a murine model of postoperative ileus obtained through intestinal manipulation (IP). In this experiment, either intragastric or intraperitoneal DMF administration prevented the delay in transit after IP, lymphocyte infiltration, activation of NF-κB and extracellular signal-regulated kinases 1/2 (Erk1/2), and it reduced IL-6 levels [63]. The anti-inflammatory effect of DMF due to impairment of NF-κB signaling was observed also in the DNBS-induced colitis model together with a substantial increase in Mn-superoxide dismutase [58]. Ethyl pyruvate, a redox analog of DMF, exerted similarly beneficial effects in a murine model of colitis, leading to the improvement of symptoms and a decrease in high mobility group box 1 (HMGB1), a key mediator of inflammation [64]. These results were also corroborated in a murine model of MS, in which ethyl fumarate reduced the number of active T cells, antigen presenting cells (APCs), and Th1/Th17-related molecules in mesenteric lymph nodes and Peyer’s patches [65]. DMF’s beneficial effects were also demonstrated in other pathological contexts featured by the presence of mycotoxins or intestinal infections. Specifically, DMF promoted the growth and morphology of the intestinal mucosa of BALB/c mice, also improving the intestinal barrier and microbiota [66]. Furthermore, DMF reduced intestinal inflammation in a mouse model of DSS-induced colitis infected with Citrobacter rodentium [67]. Despite the encouraging results in animal models, in a study by Buscarinu and colleagues [68] conducted on 25 RRMS patients, the effects of DMF on gut alterations were variable and characterized by no longitudinal pattern. 

Notably, DMF was able to protect gastric mucosa in a dose-dependent manner in a model of ethanol-induced gastric ulcers, suggesting possible applications also for the treatment of stomach disorders. The effect was similar to that exerted by omeprazole [69]. These findings were also corroborated by Sangineto and colleagues [70]. 

Table 1 reports all of the cited papers and their most relevant findings.

### 2.2. Dimethyl Fumarate’s Effects on Gut Microbiota

It is well known that, before its use as a drug in clinics, DMF was employed as a fungicide in the textile industry. Of interest, DMF is able to positively modulate the intestinal flora. One of the first pieces of evidence regarding the effects of DMF on the gut microbiota came from a study by Eppinga and colleagues [71]. Specifically, the authors demonstrated that DMF treatment in psoriasis patients was able to increase the levels of *Saccharomyces cerevisiae* in the intestine, as revealed by the analysis of fecal samples. Notably, DMF promotes the growth of this yeast also in vitro. Accordingly, DMF-related GI side effects may be induced by alterations in intestinal microbiota caused by the treatment itself. Indeed, an abnormal increase in *S. cerevisiae* levels may contribute to GI symptoms associated with DMF treatment. Yeast/fungal overgrowth can lead to infection and, in the worst cases, fungemia. Although *S. cerevisiae* is a safe yeast, its increase can generate opportunistic infections of the intestine, which are characterized by symptoms resembling those occurring after DMF treatment (e.g., abdominal pain, diarrhea, vomiting) [72,73,74,75]. The biological action of DMF on microorganisms was also demonstrated through in vitro experiments using *Clostridium perfringens*. This bacterium is associated with MS, inducing new lesions due to the tropism for the blood–brain barrier and myelin of the CNS. DMF, like other Michael acceptors (e.g., α,β-unsaturated carbonyl compounds and their derivatives), is able to inhibit C. perfringens and the release of the associated toxin [76]. Gene sequence analysis of 16 rRNA in BALB/c mice demonstrated that DMF increased the amount and diversity of the intestinal bacteria, thus promoting micro-ecologic stability as well as a positive impact on intestinal biodiversity. Notably, DMF promoted bacteria belonging to taxa associated with intestinal health, such as those producing short chain fatty acids (e.g., *Bacillus* and *Bacteroides*) [66]. In another murine model, DMF-induced gut microbiota alterations led to an increase in some bacteria (e.g., butyrate-producing *Faecalibacterium*), an effect that is associated with an improvement in memory performance due to Nrf2/ARE system induction in the hippocampus [77]. Conversely, this molecule seemed not to alter the mycobiome, although there is only one paper demonstrating no effects, however, with a limited sampling size [78]. A study conducted on 168 RRMS patients showed that DMF altered the microbiota composition: particularly, the treatment led to a decrease in *Clostridiales*, *Lachnospiraceae*, and *Veillonellacease* as well as in the phyla Firmucutes and Fusobacteria. Conversely, an increase in the phylum Bacterioidetes was detected [79]. Another study based on 36 RRMS patients showed no differences in human gut microbiota after delayed-release DMF treatment; however, a decreasing trend in Actinobacteria was observed after 2 weeks, whereas an increasing trend in Firmicutes (i.e., *Faecalibacterium*) was detected after 12 weeks [80]. The decrease in Clostridium after DMF treatment was confirmed by another study even at 6 months after drug suspension [81]. Again, DMF-related GI side effects were associated with concomitant intestinal dysbiosis. Specifically, patients that experienced GI symptoms were characterized by an increase in *Streptococcus, Haemophilus, Clostridium* and other bacteria as well as by a reduction in *Bacteroidetes*, *Akkermansia* and other Proteobacteria families [81]. Notably, DMF treatment led to a decrease in MS-associated pro-inflammatory taxa, such as *Akkermansia muciniphilia* and *Coprococcus eutactus*, and an increase in anti-inflammatory species, such as *Lactobacillus pentosus*. Interestingly, lymphopenia that developed in patients during DMF treatment was associated with the presence of *A. muciniphilia* and the concomitant absence of *Prevotella copri*, suggesting a crosstalk between these two taxa, with *P. copri* able to mitigate the effects of *A. muciniphilia* [82]. 

Table 2 reports literature regarding DMF-mediated gut microbiota alterations.

## 3. Intestinal Disorders as Targets of DMF

In accordance with the findings summarized in the previous sections, we hypothesized that DMF administration might represent a valuable approach to counteract inflammatory bowel diseases (IBDs) and celiac disease (CeD). The repurposing of DMF in the treatment of these intestinal pathologies relies on their pathogenetic mechanisms, involving cellular types in which beneficial effects of DMF have been already described.

### 3.1. Inflammatory Bowel Diseases (IBDs)

Inflammatory bowel disease is a general term that refers to two chronic and systemic disorders mainly affecting the intestine, specifically Crohn’s disease (CD) and ulcerative colitis (UC). Since 1990, the incidence of IBDs has increased worldwide, with a frequency in Western countries of 0.3% [83]. Despite the similarities in their clinical manifestations, CD and UC involve different parts of the human gut, with CD affecting intestine, stomach and/or esophagus, and UC being localized primarily in the colon. Furthermore, the histopathological hallmarks of these two disorders are different: CD is characterized by damaged mucosa, granulomas, and transmural inflammation, and UC by mucosal and submucosal inflammation, mainly localized in the crypts (e.g., cryptitis and abscesses) [84,85,86,87,88]. The differences in the clinical manifestations of CD and UC are still unclear; however, it has been hypothesized that they may rely on distinct molecular triggers (antigens/dietary particles for CD and bacteria for UC, respectively) as well as on the different immune cells recruited in the affected area (macrophages for CD and neutrophils for UC) [89]. IBD is a syndrome characterized by a plethora of symptoms ranging from abdominal pain, diarrhea, and weight loss to infiltration of neutrophils and macrophages with consequent production of pro-inflammatory cytokines [85,86]. The onset of IBDs depends on different genetic, epigenetic, and environmental factors. Genome-wide association studies (GWAS) as well as next-generation sequencing studies (NGS) led to the identification of up to 200 genes specifically associated with CD and UC. These susceptibility genes are involved in epithelial barrier homeostasis, innate and adaptive immune modulation, cell migration, autophagy, and other relevant cellular pathways [87,90,91]. Notably, both CD and UC are associated with gut microbiota alterations, which seem to contribute to the onset of IBDs [85,86]. The relevance of intestinal dysbiosis in the pathogenesis of CD and UC has been confirmed also by the potential of fecal microbiota transplantation (FMT) as a therapeutic approach [92,93,94]. 

CD and UC pathogeneses present both similarities and differences. Briefly, CD comprises a persistent immune response against luminal antigens produced by both micro- and mycobiota. One of the key leading events is the altered secretion of mucus by Paneth cells; indeed, changes in the expression of genes involved in mucus production, such as *the nucleotide-binding oligomerization domain-containing protein 2* (*NOD2*, also known *CARD15*), the *authophagy related 16 like 1* (*ATG16L1*), *the immunity related GMPase M* (*IRGM*), and the *leucine rich repeat kinase 2* (*LRRK2*) [91,92], have been reported. Polymorphisms in *NOD2* as well as in genes belonging to NF-κB promote epithelial barrier dysfunction, thus leading the luminal content to reach the *lamina propria*. Consequently, antigen presenting cells (APCs) release cytokines (e.g., IL-12, IL-23, and IFN-γ) that lead to the hyperactivation of pro-inflammatory T cells, inducing mainly a Th1 phenotype. These cytokines are also able to activate natural killer (NK) cells, which contribute to sustained intestinal inflammation. In this scenario, IL-4, IL-6 and IL-21 also play key roles in maintaining the inflammatory response [95,96]. Conversely, UC is characterized by a Th2-mediated inflammatory response after intestinal mucosal damage. Moreover, recently, a novel subset of T cells, Th9, was identified; Th9 cells are responsible for the release of IL-9, thus interfering with the mechanisms of tissue repair and intestinal barrier functionality [86]. The current IBD treatments are based on aminosalicylates, corticosteroids, and immunomodulators [97].

### 3.2. Celiac Disease

Celiac disease (CeD), also referred to as celiac sprue or gluten-sensitive enteropathy, is a serious immune-mediated condition belonging to the family of gluten-related disorders (GRDs). CeD affects genetically predisposed individuals in which the ingestion of gluten leads to several intestinal and extra-intestinal symptoms (e.g., anemia, fatigue, infertility, osteoporosis, neurologic disorders, and dermatitis herpetiformis). Currently, it is estimated that CeD affects 1% of people worldwide, although only one in 6 people are correctly diagnosed. CeD is associated with several comorbidities, and untreated or undiagnosed forms are usually characterized by severe long-term effects and consequences for the general health of diseased individuals [98]. CeD is triggered by the ingestion of gluten, a complex mixture of protein storage present in the mature seed endosperm of grains of the Triticeae group (i.e., wheat, barley and rye) and all of their hybrids (i.e., spelt, Kamut and triticale) [99]. Specifically, the pathogenic mechanism is linked to gliadin, the cytotoxic alcohol-soluble component of gluten [98]. In CD patients, ingestion of gluten leads to an abnormal response of the innate and adaptive immune system and, therefore, to the generation of an inflammatory environment characterized by the presence of lymphocyte infiltration in the epithelium and villous atrophy. The first effect of gliadin at the intestinal level is the increase in gut permeability through the zonulin release [100]. Once in the lamina propria, gluten peptides are modified by the enzymatic activity of extracellular transglutaminase 2 (TG2, TGM2), whose levels and activity are raised after tissue injury, thus increasing their binding affinity to HLA-DQ2/DQ8 molecules [101,102]. Dendritic cells (DCs) present gliadin peptides to CD4+ naïve T cells, thus promoting the repertoire expansion of gluten-specific Th1/Th17 pro-inflammatory T cells, with a concomitant production of IFN-γ, TNF-α, IL-18 and IL-21 [103,104,105]. INF-γ production increases the number of CD8+ TRCαβ+ and TCRγδ+ intraepithelial lymphocytes (IELs) with innate-like lymphokine-activated killer (LAK) activities. At the same time, intestinal epithelial cells (IECs) over-express the ligands of these receptors, respectively MICA and HLA-E, in response to the stress caused by IL-15 release. The interaction between IELs and IECs mediated by their receptors and ligands produces the release of IFN-γ and cytolysis, thus leading to tissue damage. CD4+ T cells can also interact with B cells, promoting their maturation, thus favoring the production of specific anti-TG2 and anti-gliadin antibodies [104]. Currently, the only effective therapy is a gluten-free diet (GFD) [98].

## 4. Rationale for DMF Repurposing in Intestinal Pathologies

The repurposing of DMF in the treatment of intestinal disorders firstly relies on its Nrf2-dependent mechanism of action. Indeed, Nrf2 plays a key role in enterogenesis, as demonstrated by in vivo studies [106,107]. As already reviewed by Piotrowska and colleagues [108], the Nrf2 pathway is involved in intestinal protection and in the maintenance of intestinal epithelial barrier homeostasis [109,110]. In vivo experiments demonstrate that Nrf2 knockout leads to an increased susceptibility to inflammation and oxidative stress damage in DSS-induced colitis mouse models [111]. The importance of a fine modulation of Nrf2 for gut homeostasis was confirmed by another in vivo study in which Nrf2 overexpression led to a worsening of the inflammatory status in a model of acute colitis [112]. The protective role of Nrf2 was also demonstrated in the context of CeD. Particularly, wheat germ peptides exerted a beneficial role in an in vitro model of CeD through the activation of the Nrf2 pathway [113]; moreover, resveratrol, a well-known Nrf2 inducer, led to a reduction in oxidative stress and epithelial cell damage in in vitro and in vivo CeD models [114]. Similar results were also obtained in a gliadin-induced enteropathy mouse model using conjugated linoleic acid [115]. Notably, 5-aminosalicylic acid (5-ASA), a drug used for the treatment of IBDs, in its oxidized form is able to induce the Nrf2 pathway, thus promoting an anti-inflammatory response [116]. Considering the pathogenesis of IBDs and CeD, the immunomodulatory properties of DMF (reviewed by [117]) or similar compounds might play a beneficial role by reducing inflammation, promoting a more tolerogenic phenotype in APCs [118], or inhibiting T cell maturation and activation [119]. 

Gut homeostasis is strongly affected by the enteric nervous system (ENS), in particular through the functions of enteric neurons (ENs) and enteric glial cells (EGCs). Considering the beneficial modulatory effects exerted by DMF in these two cell types (i.e., neurons and glial cells), the rationale for DMF repurposing in gut diseases may rely on enteric neuron/glia neuroprotection. As widely discussed in the literature, the ENS plays an essential role in the maintenance of intestinal health. Particularly, under physiological conditions, the crosstalk among IEC, EGCs, and ENs regulates the correct intestinal epithelial barrier function as follows: (1) EGCs secrete pro-epidermal growth factor (pro-EGF), glial-derived neurotrophic factor (GDNF), S-nitrosoglutathione (GSNO), TGF-β, 15-Hydroxyeicosatetraenoic acid (15-HETE), and 15-deoxy-delta-12,14-prostaglandin J2 (15d-PGJ2); (2) ENs release acetylcholine (ACh), vasoactive intestinal peptide (VIP), neuropeptide Y (NPY) and other factors; (3) IECs secrete mucus and antimicrobial peptides (AMPs) as protection. Inflammatory conditions or environmental stressors lead to the induction of reactive glia, with the consequent secretion of nitric oxide (NO), S100 calcium-binding protein B (S100β), nerve growth factor (NGF), IL-1β and IL-6. Under stress (i.e., inflammation, oxidative stress, infection), neurons also contribute to dysfunction of the intestinal epithelial barrier, for example, through the corticotropin-releasing factor (CRF) [120,121]. Due to the plethora of cellular functions exerted by EGCs, some disorders have been reconsidered by placing these cells at the center of attention, as in the case of CD, which can be classified as a gliopathy, as suggested by Pochard and colleagues [122]. A possible involvement of EGCs in the pathogenesis of CeD was also hypothesized by Esposito and colleagues [123], who demonstrated that CeD patients are characterized by an increased number of EGCs in the duodenum, showing an up-regulation of S100β, iNOS gene expression and NO. Furthermore, this enteric glia response, compatible with a reactive phenotype, seems to be directly induced by gliadin. 

Enteric glia may represent the principal target for DMF-mediated modulation of the ENS. Indeed, the induction of the ARE elements in the central nervous system (CNS) is restricted to astroglial cells instead of neurons [124]; furthermore, DMF leads to a decrease in inflammatory mediators after Nrf2 activation, as demonstrated in vitro and in vivo [125,126,127,128]. 

DMF-mediated modulation of the gut microbiota/mycobiota in IBDs or CeD may be attractive, considering that dysbiosis is associated with these intestinal disorders [129,130,131,132,133,134]. Beneficial effects of DMF in gut micro- and mycobiota may contribute to the resolution of dysbiosis and the rescue of taxa involved in intestinal health. Figure 3 summarizes the beneficial effects of DMF in different cellular targets in the context of intestinal disorders (e.g., CD).

## 5. Other Considerations for DMF-Based Approaches in Intestinal Disorders

All of the evidence collected in this review highlighted that DMF might be useful in treating intestinal disorders. In recent years, modulation of the Nrf2 pathway has received more and more attention, especially regarding IBDs. Different studies described the beneficial effects of Nrf2/Keap1/ARE signaling modulation in some models of gut diseases. Particularly, it has been reported that mesenchymal stem cells (MSCs) promote intestinal healing by regulating Nrf2 [135]. Notably, other molecules also exert similar effects, such as epoxymicheliolide (EMCL) [136], carnosic acid [137], melianodiol [138], miconazole [139], sulphorafane (SFN), polyphenols and proanthocyanidin extracts [140,141,142]. However, compared to all of these molecules, DMF has the advantage of being well characterized and already approved by the FDA for clinical applications. The adverse effects described for DMF (e.g., nausea, vomiting, abdominal pain, and diarrhea) are caused by the levels of methanol produced during its metabolism as well as possible alterations in gut microbiota. These symptoms are common and attributable to many intestinal diseases; considering these premises, monitoring by physicians is needed to avoid the risk of misdiagnosis. A case report by Wilkinson described a patient affected by psoriasis experiencing several GI disorders that were initially ascribed to Fumaderm^®^. The onset of microcytic anemia led to the correct diagnosis of colon cancer and partial obstruction [143], thus underling the importance of accurate patient follow-up and management, especially considering the wide spectrum of GI side effects characterizing DMF treatment. Despite these concerns, it is necessary to underline that the GI symptoms caused by DMF are reversible and can be managed through the modulation of therapeutic doses/timing of administration, without necessarily compromising its beneficial effects, as demonstrated in a case report by Kofler and colleagues [144] involving an 88-year-old woman affected by cystoid macular edema. Considering the fact that DMF is completely metabolized in the intestine, treating gut disorders using lower doses of DMF compared to those commonly used for other pathologies may represent a good strategy, since the effect would be local. In addition, a low dose of DMF may lead to a sub-toxic amount of methanol, the main cause of DMF-induced gut irritation. Another strategy might be the direct use of MMF, which is characterized by better GI tolerability and exerts protective effects against gastric ulcerations [145,146], although several studies in the literature in different contexts underline the higher efficacy of DMF compared to MMF [147,148]. DMF seems to be stable from hydrolysis at low pH. This likely means that in the stomach (~1.5/2 pH), DMF may exert its beneficial effects directly, in the absence of conversion into MMF. This aspect may represent an advantage since, as mentioned, methanol is produced after DMF catabolism. However, it is necessary to remember that, currently, the available formulations of DMF are gastro-resistant tablets. Derivatives of DMF, such as diroximel fumarate (DRF) or HYCO-3, which showed a reduced GI toxicity [149,150], might represent additional alternative strategies to avoid the onset of GI symptoms in susceptible individuals without reducing the immunomodulating and anti-inflammatory response. Notably, considering the role of trefoil factor (TFF) peptides and tight junctions (TJs) in intestinal mucosa homeostasis and damage response [151,152,153], DMF might restore the physiological levels of these molecules in GI inflammatory contexts. Particularly, the levels of TFF peptides (mainly expressed in the gastric glands and goblet cells) are dysregulated in pathological conditions: for instance, TFF3 is up-regulated in gastric ulcers, and *tff3^−/−^* mice are more susceptible to colitis after DSS administration [152]. Up-regulation of TFF3 is associated with many gastric and colon cancers; accordingly, the possible rescue of its physiological levels by DMF might also exert long-term protection, thus preventing the onset of these pathologies. Considering that Nrf2-inducing phytochemicals are able to increase the levels of hepcidin [154], DMF might also affect the expression of this important factor.

## 6. Conclusions

DMF is an effective drug already approved for psoriasis and RRMS. The growing evidence of the beneficial effects of DMF in vivo on the intestine encourages its repurposing for the treatment of GI disorders.

In conclusion, the conceptualization of novel DMF-based therapeutic approaches in intestinal diseases merits more consideration and a careful analysis by the scientific community.

## Figures and Tables

**Figure 1 ijms-24-09912-f001:**
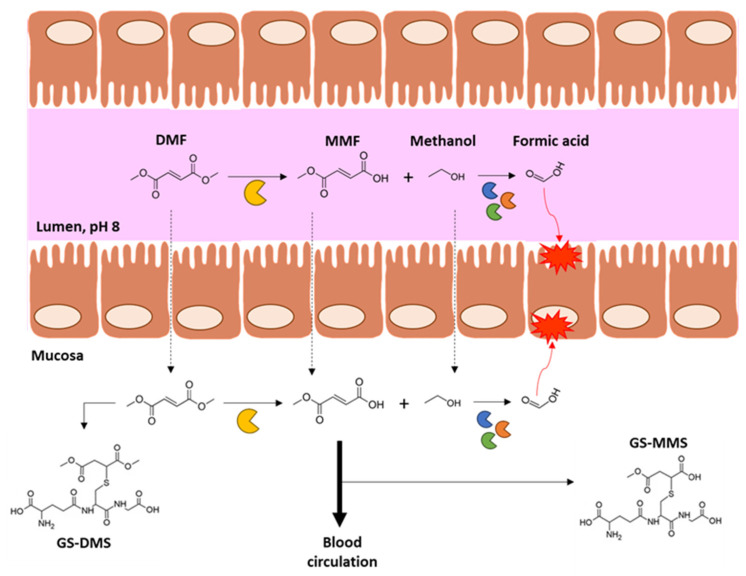
Schematic representation of DMF metabolism in the gut. Briefly, DMF is metabolized by intestinal esterases in MMF, the main DMF metabolite, and methanol, a well-known gut irritant. Methanol is converted into formic acid due to the action of several enzymes (e.g., dehydrogenases, or oxidases and dismutases). Once in the mucosa, both DMF and MMF can react with GSH to form adducts. According to the physico-chemical properties of DMF and the high presence of enzymes, this molecule is completely metabolized in the intestine. Only MMF enters in the blood circulation, from which it is then distributed to the target tissues. GS-DMS and GS-MMS indicate the glutathione adducts with DMF and MMF, respectively.

**Figure 2 ijms-24-09912-f002:**
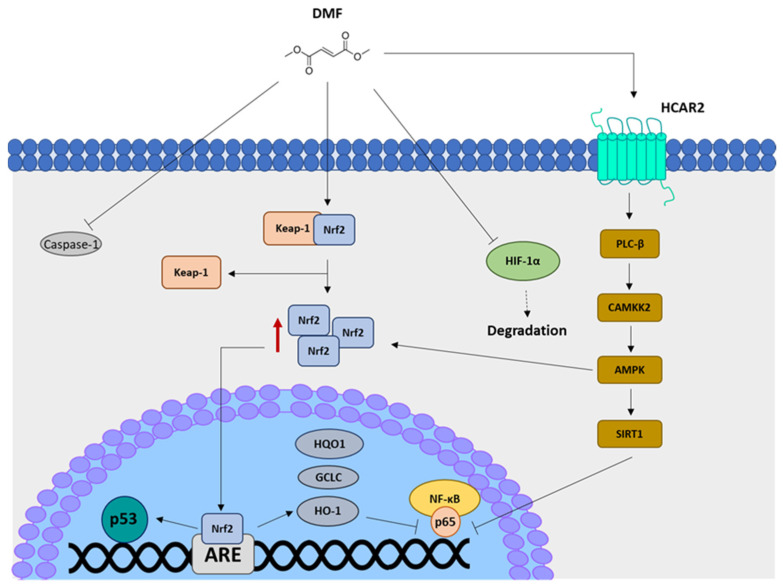
Schematic representation of DMF mechanisms of action. Briefly, once entered into the cytoplasm, DMF induces the dissociation of Keap-1 from Nrf2, thus allowing its translocation into the nucleus and the binding of target genes containing ARE sequences in their promoter (e.g., *TP53*, *HQO1*, *GCLC*, and *HO-1*). DMF can also inhibit caspase-1, thus alleviating the NLRP3-mediated inflammatory response, and HIF-1α, promoting its degradation. Another target of DMF is the NF-κB pathway, which is inhibited by the effects mediated by HO-1 or SIRT1, the latter interfering with NF-κB nuclear translocation.

**Figure 3 ijms-24-09912-f003:**
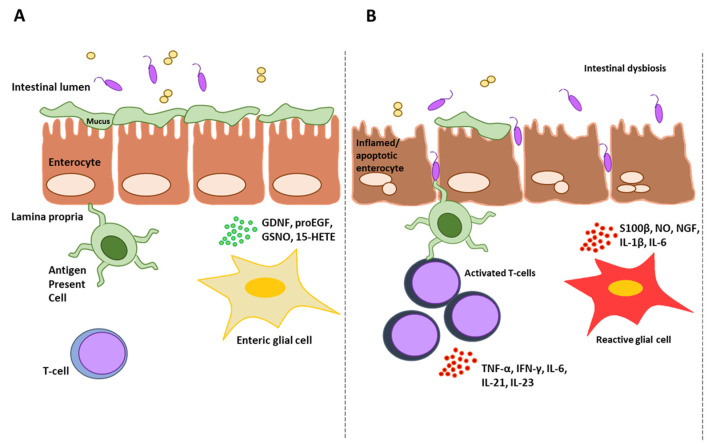
Schematic representation of DMF’s beneficial effects in the context of intestinal disorders (e.g., CD). Panels (**A**,**B**) show the physiological and pathological intestinal mucosa with the main cellular protagonists. Panel (**C**) summarizes the beneficial effects of DMF on the different cellular targets. Briefly, DMF can modulate the gut microbiota, favoring an increase in taxa associated with healthy intestinal mucosa. Moreover, DMF is also able to induce a more tolerogenic phenotype in antigen presenting cells (APCs). The anti-inflammatory effects of this molecule target enterocytes as well as T cells, whose proliferation and secretion of pro-inflammatory cytokines are inhibited. As hypothesized in the previous paragraph, DMF could exert positive effects also on the cellular components of the enteric nervous system (ENS), specifically enteric glial cells (EGCs).

**Table 1 ijms-24-09912-t001:** Published papers (in chronological order) reporting the beneficial effects of DMF and related molecules in different models of GI disorders.

Author and Year	Molecule	Type of Study	Model	Described Effects
Pereira et al., 1994 [52]	DMF	In vitro	Fisher rats (ACF model)	↓ ACF
Munday et al., 1999 [53]	DMF	In vivo	Sprague-Dawley rats (treated with naphthoquinone)	↑ DT-diaphorase
Kirlin et al., 1999 [56]	DMF	In vitro	HT-29 cells	↑ Apoptosis
Begleiter et al., 2003 [54]	DMF	In vivo	Sprague-Dawley rats (ACF model)	↑ NQO1, GST; No effects on UGT
Davé et al., 2009 [64]	EP	In vivo/in vitro	Murine IL-10 KO chronic colitis model, murine macrophages	↑ HMGB1, HO-1; ↓ RAGE, IL-12p40, NO, NF-κB DNA binding
Xie et al., 2015 [57]	DMF	In vitro	CT-26, HT-29, HCT-116, and SGC-7901 cells	↓ GSH, viability; ↑ JNK, p38, Erk1/2, autophagy markers
Liu et al., 2016 [41]	DMF	In vivo/in vitro	DSS-induced colitis model (wt or *Nrf2^−/−^* C57BL/J mice) Bone marrow cells Human THP-1 cells	↓ Body weight loss, colon shortening, MPO, iNOS, IL-1β, TNF-α, IL-6, caspase-1, NLRP3, mROS, mtDNA release; ↑ GSH, Nrf2
Casili et al., 2016 [58]	DMF	In vivo/in vitro	Experimental colitis (IL-10KO and wt mice, CD-1 mice treated with DNBS)	↓ Body weight loss, diarrhea, colon damage, MPO, TNF-α, ICAM-1; ↑ Mn-SOD, ZO-1, HO-1 impairment of NF-κB
Djedović et al., 2017 [65]	EP	In vivo	EAE mouse model	↓ T cells, APCs, Th1/Th17-related molecules in mesenteric lymph nodes and Peyer patches
Ma et al., 2017 [66]	DMF	In vivo	BALB/c mice	↑ Growth and morphology of intestinal mucosa; ↓ IEB permeability, mycotoxins
Kaluzki et al., 2019 [55]	DMF	In vitro	HT-29 and T84 cells	↓ Proliferation, cyclin D1, CDK4; ↑ p21, autophagy and apoptotic markers
Pitarokoili et al., 2019 [59]	DMF	In vivo/ex vivo	Experimental autoimmune neuritis (Lewis rats)	↓ *TLR-4* and *IFN-γ* mRNA; ↑ *Nrf2*, *HO-1*, and *FoxP3* mRNA; regulatory T cells in Peyer patches
Li et al., 2020 [62]	DMF	In vivo	DSS-induced colitis model	↓ Colon shortening, inflammation, COX-2; ↑GCLC, GPX
Van Dingenen et al., 2020 [63]	DMF	In vivo	Post-operative ileus model (C57BL/J mice under IM)	↓ Delayed transit; ↓ IL-6, lymphocyte infiltration, Erk-1/2, NF-κB
Sangineto et al., 2020 [70]	DMF	In vivo	Ethanol-induced gastric ulcer model	Protection against gut barrier dysfunction and LPS translocation
Gendy et al., 2021 [60]	DMF	In vivo	Ischemia/reperfusion model (Wistar rats)	↑ Nrf2, HO-1, Bcl-2; ↓ GSK-3β, MDA, iNOS, NF-κB, MPO, TNF-α, IL-1β, P-selectin, caspase-3
Buscarinu et al., 2021 [68]	DMF	Observational study	MS patients	Variable effects on gut barrier alterations
Patel et al., 2022 [69]	DMF	In vivo	Ethanol-induced gastric ulcer model	↓ Thiobarbituric acid reactive substance levels; protection against ulcers
Mi et al., 2023 [61]	DMF	In vivo	Necrotizing enterocolitis (C57BL/J mice treated with hypoxia and LPS)	↓ Weight loss, diarrhea, IL-6, IL-1β, TNF-α, TLR-4, NF-κB, Bax, caspase 3/9; ↑ Bcl-2
Chen et al., 2023 [67]	DMF	In vivo	DSS-induced colitis or *C. rodentium* infection	↓ Intestinal inflammation, gasdermin D-induced pyroptosis of IELs

**Table 2 ijms-24-09912-t002:** Literature studies (in chronological order) reporting the effects of DMF on gut micro- and mycobiota.

Author and Year	Molecule	Type of Study	Model	Described Effects
Eppinga et al., 2017 [71]	DMF	In vivo/in vitro	Fecal samples from psoriasis patients; *S. cerevisiae*	↑ *S. cerevisiae* in DMF-treated patients
Rumah et al., 2017 [76]	DMF	In vitro	*C. perfringens*	↓ Growth
Ma et al., 2017 [66]	DMF	In vivo	BALB/c mice	↑ Growth and morphology of intestinal mucosa; ↓ IEB permeability, mycotoxins
Katz Sand et al., 2018 [79]	DMF	Observational study	MS patients	↓ *Lachnospiraceae*, *Veillonellaceae*, Firmicutes, Fusobacteria, Clostridiales; ↑ Bacteroidetes
Storm-Larsen et al., 2019 [80]	DMF	Observational study	MS patients	Reduced trend of Actinobacteria, ↑ Firmicutes (*Faecalibacterium*)
Sadnovnikova et al., 2021 [77]	DMF	In vivo	C57BL/J mice	↑ Mitochondrial biogenesis, mitophagy, Nrf2/ARE pathway; modification of gut microbiota
Shah et al., 2021 [78]	DMF	Observational study	MS patients	No changes in gut mycobiota
Diebold et al., 2022 [82]	DMF	Observational study	MS patients	↓ *Akkermansia muciniphilia* and *Coprococcus eutactus*; ↑ *Lactobacillus pentosus*
Ferri et al., 2023 [81]	DMF	Observational study	MS patients	↓ Clostridium; ↑ *Streptococcus*, *Haemophilus*, *Clostridium*, *Lachnospira*, *Blautia*, *Subdoligranulum* and ↓ *Bacteroidetes*, *Barnesiella*, *Odoribacter*, *Akkermansia* in patients with side effects

## Data Availability

Not applicable.

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
