# Peer review of "Dimethyl Fumarate and Intestine: From Main Suspect to Potential Ally against Gut Disorders"

_ijms, 2023, doi:10.3390/ijms24129912_

Round 1

Reviewer 1 Report

The reviewer would like to appreciate the authors immense efforts in organising this review. The reviewer finds the author wrote the review in DMF on intestine and gut microbiota quite well and provided enough references. 

Having said, the reviewer would like to recommend this article for publication with a minor comment;

1. Line 41-43: Rewrite the sentences to make the reader understandable and recheck the references added to it

Author Response

Comments of REVIEWER 1

The reviewer would like to appreciate the authors immense efforts in organising this review. The reviewer finds the author wrote the review in DMF on intestine and gut microbiota quite well and provided enough references. 

Having said, the reviewer would like to recommend this article for publication with a minor comment;

  1. Line 41-43: Rewrite the sentences to make the reader understandable and recheck the references added to it

Response to Reviewer 1:

We thank the Reviewer for his/her appreciation for our review.

According to the request, we have rephrased the sentence at line 41-43, and we confirm the references reported.

Reviewer 2 Report

In my opinion it is an article that provides adequate information to the scientific community. However, I suggest putting a section on material and methods in which they detail which search engines they used, how they chose the articles they used for the review and what was the methodology to carry out the search.

 Minor editing of English language required

Author Response

Comments of REVIEWER 2

In my opinion it is an article that provides adequate information to the scientific community. However, I suggest putting a section on material and methods in which they detail which search engines they used, how they chose the articles they used for the review and what was the methodology to carry out the search.

Response to Reviewer 2:

We thank the Reviewer for his/her appreciation for our review.

According to the request, we have added a sentence (line 153-156) explaining the search and selection criteria used in this review. 

Reviewer 3 Report

Reviewer’s Comments

The manuscript “Dimethyl fumarate and intestine: from main suspect to potential ally against gut disorders” is a very interesting work. In this work, Dimethyl fumarate (DMF) is a well-characterized molecule that exhibits immuno-modulatory, anti-inflammatory, and antioxidant properties and that is currently approved for the treatment of psoriasis and multiple sclerosis. Due to its Nrf2-dependent and independent mechanism of action, DMF has a therapeutic potential much broader than expected. In this comprehensive review we discuss the state-of-art and future perspectives regarding the potential repurposing of DMF in the context of chronic inflammatory diseases of the intestine, such as inflammatory bowel disorders (i.e., Crohn’s disease and ulcerative colitis) and celiac disease. The DMF's mechanism of action, as well as an exhaustive analysis of the in vitro/in vivo evidence of its beneficial effects on the intestine and the gut microbiota, together with observational studies on multiple sclerosis patients, are here reported. While I believe this topic is of great interest to our readers, I think it needs major revision before it is ready for publication. So, I recommend this manuscript for publication with major revisions.

1. In this manuscript, the authors did not explain the disadvantages of gut disorders in the introduction part. The authors should explain the disadvantages of gut disorders.

2) Title: The title of the manuscript is not impressive. It should be modified or rewritten it.

3) Correct the following statement “Altogether, this evidence highlighted the new potential applications of this molecule in the context of inflammatory and immune-mediated intestinal diseases”.

4) Keywords: The gut disorders is missing in the keywords. So, modify the keywords.

5) Introduction part is not impressive. The references cited are very old. So, Improve it with some latest literature such as 10.3390/molecules27217368, 10.3390/molecules27207129

6) The authors should explain the following statement with recent references, “The decrease of Clostridium after DMF treatment was confirmed by another study even at 6 months after drug suspension”.

7) Add space between magnitude and unit. For example, in synthesis “21.96g” should be 21.96 g. Make the corrections throughout the manuscript regarding values and units.

8) The author should provide reason about this statement “Despite the similarities in their clinical manifestations, CD and UC involve different parts of the human gut, with CD affecting intestine, stomach and/or esophagus, and UC being localized primarily in the colon”.

9. Comparison of the present results with other similar findings in the literature should be discussed in more detail. This is necessary in order to place this work together with other work in the field and to give more credibility to the present results.

10) Conclusion part is very long. Make it brief and improve by adding the results of your studies.

11) There are many grammatic mistakes. Improve the English grammar of the manuscript.

Minor editing of English language required

Author Response

Comments of REVIEWER 3

The manuscript “Dimethyl fumarate and intestine: from main suspect to potential ally against gut disorders” is a very interesting work. In this work, Dimethyl fumarate (DMF) is a well-characterized molecule that exhibits immuno-modulatory, anti-inflammatory, and antioxidant properties and that is currently approved for the treatment of psoriasis and multiple sclerosis. Due to its Nrf2-dependent and independent mechanism of action, DMF has a therapeutic potential much broader than expected. In this comprehensive review we discuss the state-of-art and future perspectives regarding the potential repurposing of DMF in the context of chronic inflammatory diseases of the intestine, such as inflammatory bowel disorders (i.e., Crohn’s disease and ulcerative colitis) and celiac disease. The DMF's mechanism of action, as well as an exhaustive analysis of the in vitro/in vivo evidence of its beneficial effects on the intestine and the gut microbiota, together with observational studies on multiple sclerosis patients, are here reported. While I believe this topic is of great interest to our readers, I think it needs major revision before it is ready for publication. So, I recommend this manuscript for publication with major revisions.

  • Response to Reviewer 3:

We thank the Reviewer for his/her comments and suggestions to our review.

Reviewer: 1. In this manuscript, the authors did not explain the disadvantages of gut disorders in the introduction part. The authors should explain the disadvantages of gut disorders.

- Response: According to the request, we have mentioned the general disadvantages of gut disorders (i.e., socio-economic burden and impact in patients’ quality of life) in line 143-145. However, we point out that more detailed symptoms, for each disease, are described in the following specific sections.

2) Title: The title of the manuscript is not impressive. It should be modified or rewritten it.

- Response: We thank the Reviewer for this suggestion. We slightly modified the title making it more impressive: “Dimethyl fumarate and intestine: overturn from main suspect to potential ally against gut disorders”

3) Correct the following statement “Altogether, this evidence highlighted the new potential applications of this molecule in the context of inflammatory and immune-mediated intestinal diseases”.

- Response: According to the Reviewer’s request, we corrected the sentence as follows: “Based on the collected evidence, we highlight the new potential applications of this molecule in the context of inflammatory and immune-mediated intestinal diseases”

4) Keywords: The gut disorders is missing in the keywords. So, modify the keywords.

- Response: We thank the Referee for this useful suggestion; “gut disorders” is now included in the keywords.

5) Introduction part is not impressive. The references cited are very old. So, Improve it with some latest literature such as 10.3390/molecules27217368, 10.3390/molecules27207129

- Response: We respect the opinion of the Reviewer regarding our Introduction, although we partially disagree with the statement “The references cited are very old”, since 30 over 70 references are articles published in the last 5-6 years. We are aware that some references reported in the Introduction are old, but this is due to the fact that the main information regarding DMF’s characterization, mechanism of action (e.g. Nrf2 pathway) and effects on the intestinal mucosa, has been collected many years ago. We appreciate that Reviewer suggested two papers to be cited, but we cannot use them, since they are completely out of topic, as clearly displayed by the titles reported below:

10.3390/molecules27217368: Synthesis, In Vitro Biological Evaluation and In Silico Molecular Docking Studies of Indole Based Thiadiazole Derivatives as Dual Inhibitor of Acetylcholinesterase and Butyrylchloinesterase”

10.3390/molecules27207129: “Synthesis, In Vitro Anti-Microbial Analysis and Molecular Docking Study of Aliphatic Hydrazide-Based Benzene Sulphonamide Derivatives as Potent Inhibitors of α-Glucosidase and Urease”).

6) The authors should explain the following statement with recent references, “The decrease of Clostridium after DMF treatment was confirmed by another study even at 6 months after drug suspension”.

- Response: We thank the Reviewer for this comment. The missing reference, that was previously cited below in the text, is now appropriately placed.

7) Add space between magnitude and unit. For example, in synthesis “21.96g” should be 21.96 g. Make the corrections throughout the manuscript regarding values and units.

- Response: Thanks to the Reviewer for this punctual observation. We have added all the spaces between magnitude and unit along the manuscript.

8) The author should provide reason about this statement “Despite the similarities in their clinical manifestations, CD and UC involve different parts of the human gut, with CD affecting intestine, stomach and/or esophagus, and UC being localized primarily in the colon”.

- Response: Although displaying similar clinical symptoms, CD and UC actually involve different parts of the GI tract; the reason of this discrepancy is still unclear and debated by the global scientific community. However, in a Letter to the Editor of 2013 published by the World Journal of Gastroenterology Pathophysiology, Qin X. suggested that the differences of CD and UC localization may rely on the different triggers (antigens/dietary particles for CD, and bacteria for UC) as well as the immune cells recruited in the affected area. We have now reported this reference in the manuscript.

  1. Comparison of the present results with other similar findings in the literature should be discussed in more detail. This is necessary in order to place this work together with other work in the field and to give more credibility to the present results.

- Response:  Thanks to the Reviewer for his/her comment, that however seems to be not applicable to our manuscript. Indeed, this is the first review article focusing on the repurposing of DMF in gut disorders, mentioning all the research articles in support of our hypothesis. Other reviews proposing DMF for the treatment of other pathologies besides MS and psoriasis (e.g., cancer, cardiovascular and ocular disorders) have been published recently and cited in the current manuscript.

10) Conclusion part is very long. Make it brief and improve by adding the results of your studies.

- Response: We kindly point out that our manuscript is actually a review article thus the request of “adding results” is not affordable. Since the Conclusion (paragraph 6) counts only 5-6 lines (57 words), we think that making it shorter is not practical.

11) There are many grammatic mistakes. Improve the English grammar of the manuscript.

- Response: English editing and revision of the manuscript has been performed.

Reviewer 4 Report

See attached file

Author Response

Comments of REVIEWER 4

The paper is well written and keeps the reader stimulated to think on the topic. The sections are in a guiding seqåuence and the figures are helpful tools. However, I like to stimulate the authors to consider some remarks:

  1. There is a quite high number of reviews regarding DMF published recently (s. examples below). Please consider to incorporate at least some of them, in particular those which cover different aspects in comparison to the present manuscript.

- Response: Thanks to the Referee for his/her suggestions. We have cited some of the suggested reviews, selecting those most in line with our topic.

  1. I like to ask for a precise comment:
  2. On the fate of DMF in the gastric – acidic – environment.

- Response: We thank the Reviewer for this interesting question. As already reported in the manuscript (line 61-62, ref. 16), “DMF is preferentially hydrolyzed at alkaline pH instead of acidic one”. This likely means that in the stomach, which is characterized by a ~1.5/2 pH, DMF might exert its beneficial effects directly, in the absence of conversion in MMF. This aspect may represent an advantage since methanol is produced after DMF catabolism. However, it is necessary to remember that, currently, the available formulations of DMF are gastro-resistant tablets.

  1. Is there a difference between the effects, metabolism respectively of DMF in healthy individuals compared to those suffering by gastric ulcers? A damage in gastric mucosa can be associated with a “leaky” condition of gastric wall. What does that mean for DMF?

- Response: Thanks to the Referee for his/her observation. To our knowledge, there are no papers describing differential effects of DMF in healthy individuals compared to those affected by gastric ulcers. The “leaky” condition might benefit from DMF treatment since, as reported for intestine, this molecule is able to rescue the epithelial barrier functionality. We have reported the findings on gastric ulcers in the manuscript, as reported in line 228-231.  

  1. A “leaky” gut is a common phenomenon (hunger, parasites, inflammation etc). Please give a comment about the interaction of this condition with the discussed substance.

- Response: We thank the Reviewer for this interesting request. As reported in our Review, DMF exerts beneficial effects in counteracting increased intestinal permeability (i.e., “leaky gut”) in the context of IBD and celiac disease. Similarly, DMF might rescue this gut dysfunction present also in other conditions, such as infestations. In this specific condition, it is necessary to unveil the direct effects of DMF on common intestinal parasites. Currently, there are no papers on this topic.

  1. Butyrate is a key-compound to maintain a productive and beneficial microbiome and the gut barrier competence. Is there an effect of DMF on butyrate producing bacteria?

- Response:  Thanks to the Referee for the interesting question. As reported, nowadays there are few papers that describe the effects of DMF on gut microbiota. As already reported in our manuscript, DMF is able to increase the levels of Faecalibacterium, a class of butyrate-producing bacteria [Ref. 77, 80]. We have specified this information at line 259.

  1. I think it could be beneficial to consider some further interactions with
  2. Trefoil factor – contributing to mucosal barrier
  3. Proteins profiling the tight junctions – but partially lost during dysbiosis in the intestine
  4. Hepcidin – regulating Fe-uptake

- Response:  We thank the Reviewer for these interesting ideas. Unfortunately, despite DMF effects on these proteins might be hypothesized, there are no data in literature linking this molecule to the abovementioned factors. However, we have proposed a hypothesis at line 474-484.

  1. Please include an comment on the interaction of DMF with cell maturation and apoptosis (s. lit).

- Response: We agree that the point raised by the Review is important, indeed we have mentioned the apoptotic affects exerted by DMF in HT-29 cells, used as model of intestinal carcinoma [Kirlin et al., 1999].

  1. The authors mention a “detoxifying” capability of DMF. I think this is highly relevant for public health referring humans and farm animals. I recommend to give more information of this aspect (s. literature below)

- Response: We thank the Referee for his/her observation. The term “detoxifying” does not refer to the detoxification of xenobiotics but to the antioxidant/protective activities exerted by the Nrf2-target enzymes. For example, HO-1 is implicated is oxidative stress response and in heme detoxification. In our opinion, further discussing this argument here, in the present review, may be misleading to the readers and not appropriate considering the main topic.

  1. It would be great to have a statement on the perspectives of the drug to modify to the loss in neuronal functions in cases of Morbus Huntigton and/or Alzheimer’s Disease.

- Response: Thanks to the Reviewer for the important suggestion. Neurodegenerative disorders are characterized by GI symptoms due to enteric nervous system (ENS) dysfunction (PMID:29360466) as well as microbiota alterations (PMID:35237258). However, in our opinion, this topic requires a long dissertation since it needs supplementary considerations/comments based on the pathogenesis of these disorders and the involvement of severe alterations of the Central Nervous System (CNS). As reported by the Referee, there are some evidence in the literature of the DMF efficacy in the context of neurodegenerative disorders (NDDs). Consideration of DMF repurposing for the treatment of ENS dysfunction in NDDs should take into account also the effects exerted in the CNS. In our opinion, this topic would need to be investigated properly and in detail in a separate review. We are already working in a new manuscript describing all the pre-clinical data collected on DMF in different disorders, including NDDs, thus we prefer not discussing here this point.   
